# GENOMEOCEAN: EFFICIENT FOUNDATION MODEL FOR GENOME GENERATION

## ABSTRACT

We introduce GenomeOcean, a 4-billion-parameter genome foundation model that natively generates DNA sequences that are adherent to the input context. With an efficiency-oriented model design, GenomeOcean is 80 times faster than existing models of similar size in genome generation. Unlike most existing genome foundation models—such as DNABERT and Nucleotide Transformers—that are designed for discriminative tasks, GenomeOcean leverages generative modeling to unlock new potentials in genomics research. Diverging from the traditional reliance on reference genomes—which possess inherent biases—GenomeOcean is exclusively trained on large-scale curated environmental samples collected from diverse ecosystems, including oceans, lakes, forests, and soils. This extensive genomic diversity, encompassing uncultured and uncharacterized organisms, allows GenomeOcean to generate sequences that better reflect the true diversity of life. In a series of automated evaluations, we demonstrate GenomeOcean's capability to understand and follow context sequences. Compared to existing models, GenomeOcean not only better retains species information but also produces sequences with more appropriate open reading frame lengths and codon usage bias. We anticipate the open release of GenomeOcean to open up new possibilities in genomics and computational biology research. [1]

## 1 INTRODUCTION

The rapid advancement of genome sequencing technologies has triggered an explosion of genomic data, offering unprecedented opportunities to explore life's molecular intricacies (Rhoads & Au, 2015; Hu et al., 2021). Effectively analyzing and interpreting this vast data requires sophisticated computational models capable of uncovering previously unattainable patterns and insights. In this context, large language models (LLMs), particularly genome foundation models (Ji et al., 2021; Dalla-Torre et al., 2023; Nguyen et al., 2023; Zhou et al., 2023; Schiff et al., 2024), have emerged as powerful tools in genomics.

Genome foundation models treat DNA as a language composed of 4 nucleotide bases. These models have outperformed traditional methods in various discriminative tasks such as promoter prediction and splice site detection (Le

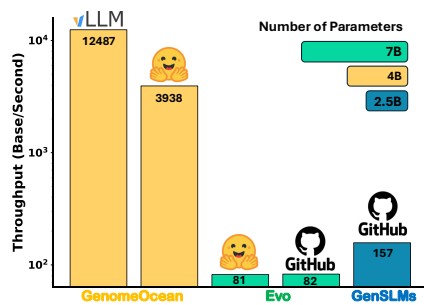

Figure 1: Genome generation throughput measured by base-pairs per second with 1000-bp prompt on a single NVIDIA A100 80GB GPU.

et al., 2022; Zhang et al., 2022; Wang et al., 2022; Lee et al., 2022). By learning contextual representations of genomic sequences, they have enhanced our ability to predict functional elements and understand gene expression (Avsec et al., 2021; Novakovsky et al., 2023). However, most genome foundation models focus on discriminative tasks, leaving the potential of generative models in genomics largely unexplored. Generative genome models hold the promise of synthesizing new DNA sequences, which could be invaluable in synthetic biology and in designing organisms with desired traits. For a generative genome model to be valuable and accessible in real-world applications, it should meet two fundamental criteria: contextual adherence and computational efficiency.

---

[1]Model, codes, and data will be publicly available.

On the one hand, the model should be contextual adherent. Besides ensuring the generated sequences are biologically plausible, they should faithfully follow the input context (e.g., the given DNA sequence) instead of producing irrelevant sequences. For instance, the generated sequence should retain the same species-specific information and demonstrate appropriate functional characteristics. This context awareness is crucial for maintaining biological relevance and applicability in downstream analyses. On the other hand, computational efficiency is essential. Generating novel, realistic, and biologically valid DNA sequences often requires extensive experimentation. For example, in the study of IS200/IS605 elements, Nguyen et al. (2024) generated over one million candidates using a large pool of hyperparameters. Efficiency, therefore, plays a key role in enabling real-world-scale studies and accelerating the iterative experiments that are common in this area of research.

To encourage contextual adherence, the diversity of training data is crucial. By learning from varied genomic contexts, the model can distinguish underlying patterns and generate sequences accordingly. Thus, unlike existing models that largely rely on reference genomes of selected species, we train GenomeOcean exclusively on a large set of curated environmental samples from diverse ecosystems. Environmental samples provide a more comprehensive representation of Earth's genetic diversity, allowing our model to learn from a vastly larger and more varied genetic repertoire. These microorganisms represent the largest reservoir of genetic and functional diversity on our planet. By training on this diverse data, GenomeOcean can differentiate closely related species based on subtle genetic features and synthesize artificial genomes that reflect this fine-grained diversity.

Furthermore, to make an informative model design that encompasses both expressiveness and efficiency, we conduct a series of preliminary experiments on existing technologies in the genome foundation model and large language models, including tokenization, model architecture, and training objectives. Based on those empirical insights, we design GenomeOcean by adapting and integrating the most suitable techniques. GenomeOcean is built upon a Transformer Decoder architecture (Vaswani et al., 2017) that integrates a series of efficiency-oriented techniques, including Group-Query Attention (GQA) (Ainslie et al., 2023), FlashAttention-2 (Dao, 2023), and vLLM (Kwon et al., 2023). Besides the model architecture, we identify the importance of the tokenizer's selection when building a billion-parameter genome foundation model. For example, the compactness of the tokenizer plays a large role in the model's inference throughput. As shown in Figure 1, GenomeOcean achieves 50 times higher throughput than Evo with the same HuggingFace (Wolf et al., 2020) inference framework, and this efficiency improvement is largely attributed to the more compact input sequence. We detail the reasoning and empirical results behind our model design in Section 2.

Given the absence of a standardized evaluation method for genome generation, we developed a suite of automated experiments to compare GenomeOcean with existing models. Our evaluations assess the coherence of the generated sequences to their input context, along with their similarity to ground truth data at both the distributional and individual levels. Results show that GenomeOcean generates sequences with greater context awareness, including more species-specific information, appropriate open reading frame lengths, and better estimation of codon usage bias. These findings underscore its capability to produce biologically plausible and contextually relevant genome sequences.

## 2 Background and Preliminary Experiments

Deoxyribonucleic acid (DNA) is a molecular structure composed of two intertwined strands forming a double helix. Each strand is made up of four fundamental building blocks known as nucleotides: adenine (A), thymine (T), cytosine (C), and guanine (G), which pair complementarily across the strands. Genome foundation models (Ji et al., 2021; Dalla-Torre et al., 2023; Nguyen et al., 2023; Zhou et al., 2023; Schiff et al., 2024; Nguyen et al., 2024; Zvyagin et al., 2022) treat DNA sequences as text sequences with just four unique characters, applying large language model (LLM) techniques to analyze them. There are three main design spaces of genome foundation models: *tokenization*, *training objective*, and *model architecture*. As existing models are often pre-trained on different datasets and use varied combinations of components, directly comparing their performance does not necessarily reveal the real impact of each component. To inform the architecture design of GenomeOcean, we conducted a series of preliminary experiments to assess each component fairly. In this section, we review existing works and provide new empirical results on genome foundation models. We discuss tokenization in Section 2.1. As training objectives are highly related to the model architecture, we discuss them simultaneously in Section 2.2.

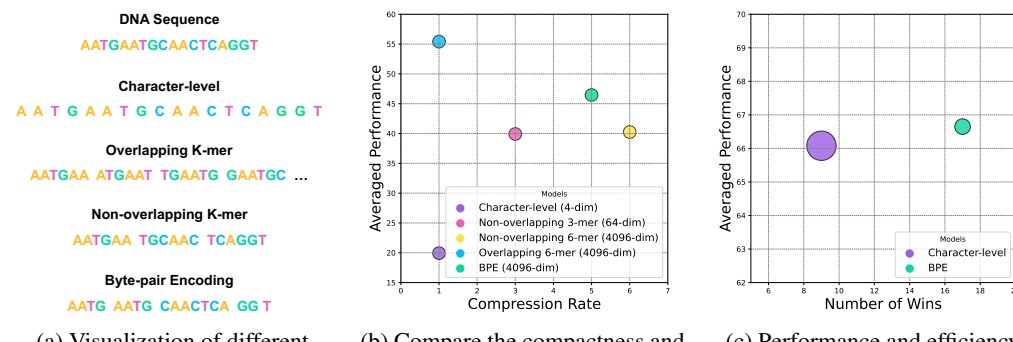

(a) Visualization of different tokenizer.

(b) Compare the compactness and expressiveness of DNA tokenizers.

(c) Performance and efficiency with char and BPE tokenizers.

Figure 2: Visualization and empirical results of each tokenizer in the context of genome modeling.

## 2.1 TOKENIZATION

Tokenization serves as the first step of vectorizing a DNA sequence. Figure 2a visualizes different DNA tokenizers. DNABERT (Ji et al., 2021) uses a commonly applied method in genomics called overlapping k-mer tokenization. This technique converts a DNA sequence into a series of tokens by scanning the sequence with a fixed-size sliding window, typically with a stride of 1. However, as highlighted by Zhou et al. (2023), this method suffers from information leakage in the context of language modeling, as overlapping tokens share a significant portion of identical characters. To address this, they propose the Byte-Pair-Encoding (BPE) (Sennrich et al., 2016) tokenization for DNA sequence, which compresses a DNA sequence into a set of non-overlapping high-frequency fragments. Moreover, non-overlapping k-mer tokenization is explored in Nucleotide Transformers (Dalla-Torre et al., 2023) and GenSLMs (Zvyagin et al., 2022), where sequences are divided into non-overlapping tokens of length $k$. BPE and non-overlapping k-mer naturally avoid the information leakage issue in overlapping k-mer tokenization while effectively reducing the input sequence length. To achieve base-level resolution, character-level tokenization, which splits each sequence by characters, is employed by HyenaDNA (Nguyen et al., 2023) and Caduceus (Schiff et al., 2024).

We evaluate different tokenization methods based on two criteria: compactness and expressiveness. Compactness refers to the method's ability to compress the input sequence, which is important given that computational efficiency depends heavily on input sequence length as foundation models scale. Expressiveness measures how well the tokenized sequence captures the necessary information, helping the model to understand the context and generate accurate sequences. To measure the compactness, we use the compression rate, which indicates how many times the tokenizer reduces the sequence length. For example, the compression rate is 5 if the DNA sequence has 100 base pairs and the tokenized sequence contains 20 tokens. To evaluate the expressiveness, we view each tokenizer as a DNA feature extractor and use the discriminativeness of the feature as the estimation of its expressiveness. Specifically, for an input DNA sequence, we respectively tokenize it with a tokenizer and use the token frequency as the feature of the DNA sequence. We assess the features on the GUE benchmark (Zhou et al., 2023), which includes 28 genome classification datasets covering both mammalian and microbial genomes. We train a multi-layer perceptron (MLP) on the features using the training data of each dataset and evaluate it on the test set.

Details results are presented in Section A.2 in the Appendix. Figure 2b summarized the average performance of the tokenizers on the GUE benchmark. The results show that the overlapping 6-mer tokenizer has the highest expressiveness, aligning with its frequent use as a feature extractor in genomics. However, its information leakage makes it unsuitable for language modeling tasks. BPE, which has similar compression rates to non-overlapping k-mer tokenization, demonstrates better expressiveness, likely because it learns high-frequency tokens from the corpus rather than relying on all possible $k$-mer combinations. In contrast, character-level tokenization performs worse, likely since its too few feature dimensions impair the expressiveness.

As the previous experiments may be unfair to the character-level tokenizer, we conduct further experiments to compare it with BPE, the winning candidate of the first experiment. Specifically, we sought to understand whether the base-level resolution provided by the character-level tokenizer justifies the increased computational complexity from longer input sequences. To do so, we train two masked language models from scratch, one using a BPE tokenizer and the other a character-

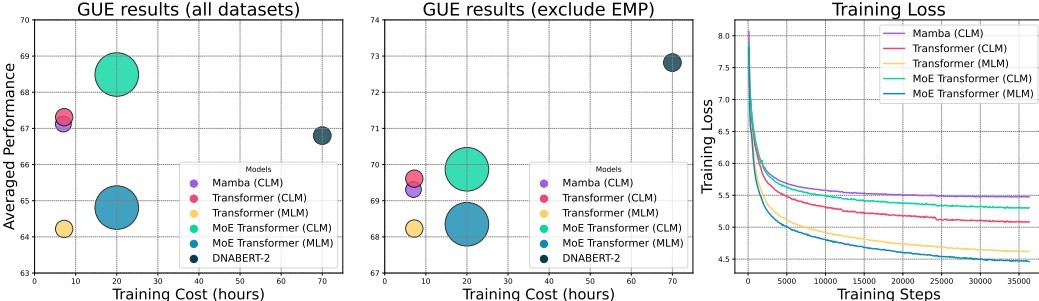

Figure 3: Preliminary experiments on different model architecture and training objectives. We compare them by the pre-training loss and averaged performance on 28 downstream datasets.

level tokenizer, keeping the architecture, hyperparameters, and training data identical. We use the BPE tokenizer from DNABERT-2 and the character-level tokenizer from HyenaDNA, following DNABERT-2's model architecture and pre-training dataset. After pre-training, we fine-tune both models on the GUE benchmark and compare their performance using average performance and the number of datasets where each model performs better. As shown in Figure 2, the model trained with the character-level tokenizer performs slightly worse than the BPE-based model, despite requiring six times more FLOPs to process a 1000-base pair sequence. The increased sequence length could significantly impair a model's training and inference efficiency. Consequently, we select BPE as the tokenizer for GenomeOcean.

## 2.2 MODEL ARCHITECTURE AND TRAINING OBJECTIVE

In genome modeling, two primary types of architectures are commonly employed: Transformers (Vaswani et al., 2017) and State Space Models (SSMs) (Gu & Dao, 2024). Models such as DNABERT, DNABERT-2, Nucleotide Transformers, and GenSLMs utilize Transformer-based architectures, while HyenaDNA, Caduceus, and Evo adopt SSM architectures. Both Transformer and SSM architectures were considered as candidates for GenomeOcean. Additionally, Mixture-of-Experts (MoE) models (Rajbhandari et al., 2022; Jiang et al., 2024), which have shown promising performance in language modeling, were also explored. MoE models contain a large number of parameters, but only a small subset is activated during inference. This allows them to retain strong representational capabilities while remaining computationally efficient through sparse activation. Given these advantages, we also investigate their applicability to genome modeling. Most existing genome models are trained with either a BERT-style (Devlin et al., 2018) masked language modeling (MLM) objective (Ji et al., 2021; Zhou et al., 2023; Dalla-Torre et al., 2023; Schiff et al., 2024), or a GPT-style (Radford et al., 2019) causal language modeling (CLM) objective (Zvyagin et al., 2022; Nguyen et al., 2023; 2024). Since GenomeOcean is designed for genome generation, CLM is the natural choice. However, we wanted to rigorously assess the relative effectiveness of these training objectives in genome modeling.

To ensure a fair comparison of different architectures, we selected three representative models. Mamba (Gu & Dao, 2024) was chosen to represent SSMs, Mistral (Jiang et al., 2023) for dense Transformers, and Mixtral (Jiang et al., 2024) for Transformers with MoE. For Mamba, which is a unidirectional model, we used causal language modeling. For the dense and MoE-based Transformers, we made them unidirectional and bidirectional by adjusting attention masks and trained them with both causal and masked language modeling, respectively. All models used the same BPE tokenizer and pre-training corpus from DNABERT-2 and were trained with identical setups for 3 epochs. The specific hyperparameters are detailed in Table 5 in the Appendix. To maintain similar computational efficiency (measured as the time taken per training step) across models, we adjusted the hidden size and number of layers for the Mamba model to match its dense Transformer counterpart. For the MoE-based Transformer, we kept most hyperparaters the same as the dense Transformer but increased the number of experts to 8, activating 2 experts during both pre-training and fine-tuning. The models were compared based on pre-training time (using 8 NVIDIA A100 80GB GPUs), training loss, and their average performance on the GUE benchmark. Given that unidirectional models perform significantly better on Epigenetic Marks Prediction (EMP) tasks, which could bias the overall benchmark results, we also exclude the EMP datasets and compare models in the rest ones. We further compared these models against the official DNABERT-2 checkpoint to validate our pre-training.

Figure 3 illustrates the performance of all models. Each circle represents a model, with the circle size corresponding to the model's relative number of parameters. While BERT-style bidirectional models are typically better suited for discriminative tasks, we found no significant benefit from using

bidirectional models in the context of genome foundation models. In both dense and MoE architectures, Transformer models with causal language modeling achieved slightly better performance, suggesting that generative pre-training better encourages the discovery of underlying patterns in genome sequences. When compared to a Transformer model with similar computational efficiency and the same training objective, Mamba exhibited slightly worse performance in both pre-training and fine-tuning phases. Furthermore, while MoE architectures consistently improved training loss and fine-tuning results for both masked and causal language modeling, the improvements were marginal considering the associated costs. Specifically, MoE training incurred a threefold increase in pre-training time, a twofold increase in active parameters, a sevenfold increase in total parameters, and significantly higher resource requirements when scaling to (tens of) billions of parameters.

## 3 MODEL

In this section, we introduce the architecture and implementation of GenomeOcean.

### 3.1 MODEL ARCHITECTURE

Based on the above findings, we implement GenomeOcean using an optimized Transformer Decoder architecture (Vaswani et al., 2017) with the causal language modeling objective. We leverage several recent techniques to improve the efficiency and representation capability.

**Group-Query Attention** (Ainslie et al., 2023). To alleviate the memory bandwidth demands of the Transformer model, we substitute the standard multi-head attention with group-query attention (GQA). GQA reduces the number of key and value heads to improve the inference scalability.

**FlashAttention-2** (Dao, 2023). We employ FlashAttention-2 to address the memory and computational inefficiencies of the vanilla attention implementation. FlashAttention-2 optimizes the attention mechanism by leveraging the GPU's memory hierarchy. This approach accelerates the attention computation while preserving exact attention results.

We also replace the ReLU activation function and standard layer normalization with **Sigmoid Linear Unit (SiLU)** activation function Elfwing et al. (2018) and **RMSnorm** (Zhang & Sennrich, 2019) to stabilize model training and improve the model's representation capability. We adapt **Rotary Positional Embedding** (Su et al., 2023) for better positional representation and more flexibility in length extrapolation. GenomeOcean has 4 billion parameters. It contains 24 Transformer layers with 3072 hidden size, 16384 intermediate size, 12 query attention heads, and 4 key-value attention heads.

### 3.2 IMPLEMENTATION

GenomeOcean is pre-trained on curated environmental samples from a variety of ecosystems, including lakes, oceans, and forests. After removing the low-quality and duplicated sequences, we achieve a pre-training dataset with around 700 billion base pairs. We train a BPE tokenizer with 4096 tokens (including special tokens) on this pre-training dataset as the tokenizer of GenomeOcean. The pre-training of GenomeOcean consists of two stages. In the first stage, we train it with a max sequence length of 1024 tokens, a batch size of 4 million tokens, and a peak learning rate of $4e\text{-}4$. The learning rate linearly increases from 0 to $4e\text{-}4$ in the first 2000 steps and decreases to $4e\text{-}5$ at the end of training. We train GenomeOcean for 59000 steps in the first stage, which is equivalent to 1.8 epochs on the training data. In the second stage, we increase the max sequence length to 10240 tokens, keep the batch size of 4 million tokens, and use a learning rate of $1e\text{-}4$. We train GenomeOcean for 1600 steps in the second stage. As a result, the max sequence length of GenomeOcean is 10240 tokens, which is equivalent to around 51000 base pairs. This max sequence length can be further extended by tens of times through interpolations (Peng et al., 2023). We leave this to future versions as the current context length is enough for most tasks. On 64 NVIDIA A100 GPUs across 16 compute nodes, the first stage costs 14 days, and the second stage costs 1 day. We implement efficient multi-node training with DeepSpeed (Rajbhandari et al., 2020). In inference, we deploy GenomeOcean to **vLLM** (Kwon et al., 2023), a framework that optimizes memory usage and increases the throughput for large language models (LLMs) inference. It uses PagedAttention to reduce memory waste, share memory across requests, and improve inference speed. As shown in Figure 1, GenomeOcean achieves 3× more throughput with vLLM compared to the HuggingFace implementation.

## 4 EXPERIMENTS

In this section, we conduct empirical analyses on the models' genome generation capability. Quantifying the *goodness* of a generated genome sequence is a vastly understudied problem. Most metrics

commonly used in natural language generation do not apply to this domain. For example, matching-based metrics such as BLEU (Papineni et al., 2002) are ineffective for genome evaluation. Due to the immense genomic diversity in nature and the absence of strict grammar and syntax rules in genome sequences (with the exception of known functional regions like coding sequences or regulatory elements), discrepancies between a generated sequence and a reference sequence do not necessarily imply errors. Besides, human evaluations (Ouyang et al., 2022) are not suitable for generated genome sequences. Genome sequences are not readily interpretable by non-experts; only specialists equipped with professional tools can evaluate generated sequences from several perspectives, such as structural validity, functional plausibility, adherence to known biological patterns, and the presence of conserved motifs. However, even these expert evaluations may not be fully conclusive.

Therefore, we design a suite of automated evaluations. We assess the generated sequences (*i.e., outputs*) along two dimensions: 1) adherence to the context sequence (*i.e., input*), and 2) similarity to ground-truth sequences (*i.e., real sequences following the input*). The first dimension ensures that the model maintains genomic coherence and produces contextually appropriate output, while the second dimension measures the model's ability to capture the statistical and functional properties of real genome sequences. To examine the adherence of the generated sequence to the prompt, we employ existing discriminative genome foundation models (e.g., Nucleotide Transformers) as the judges. To estimate the similarity between the ground-truth and generated sequences, we compared the sequences on biological properties, including codon usage bias and the lengths of open reading frames (ORFs). We present the experiment on context adherence in Section 4.1 and the experiments on ground-truth similarity in Section 4.2. We compare the generated sequences with real data at both the distributional level and the individual sequence level. At the distributional level, we examine whether the models can understand the underlying distributions of input data and generate sequences with matching distributions. At the individual sequence level, we perform pairwise comparisons between each generated sequence and its corresponding context or ground-truth sequence to assess the generation at a finer granularity. Evaluation metrics are selected task by task.

**Baselines.** We compare GenomeOcean with state-of-the-art generative genome foundation models Evo and GenSLMs. **Evo** (Nguyen et al., 2024) is a generative genome foundation model contains 7 billion parameters that are trained on the OpenGenome dataset with 300 billion nucleotide bases. It is built on the StripedHyena architecture, which hybridizes attention and hyena operators. **GenSLMs** (Zvyagin et al., 2022) is a collection of generative genome foundation models ranging from 25M, 250M, 2.5B, to 25B parameters, which are originally designed to learn the evolutionary landscape of SARS-CoV-2 genomes. It is trained on over 110 million prokaryotic gene sequences with causal language modeling. As shown in Figure 1, GenSLMs suffer from low generation throughput. In our preliminary experiments, generating all the sequences required for our evaluation pipeline took over 30 days on a single NVIDIA A100 GPU when using the 25B parameter GenSLMs model. Due to the immense computational cost of this model, we only compare it with the second-largest GenSLMs model with 2.5B parameters. Overall, the number of parameters and the size of pre-training datasets for Evo, GenSLMs, and GenomeOcean are comparable. For all the models, we generate one sequence per input. For baselines, we use their default inference hyperparameters from their official GitHub repositories. Evo uses a temperature of 1.0 and a top-p of 1.0, while GenSLMs uses a temperature of 1.0 and a top-p of 0.95. For a fair comparison, we maintain the same hyperparameters for GenomeOcean in all tasks, including a temperature of 1.0 and a top-p of 0.95.

## 4.1 Adherence to Context

In this section, we examine how well the generated sequence adherent to the given context. As existing generative genome foundation models are all pre-trained on microbial genomes, where huge amount of distinct species exist, we evaluate this by validating whether the generated sequences from each model retains the species-related characteristics of the given context.

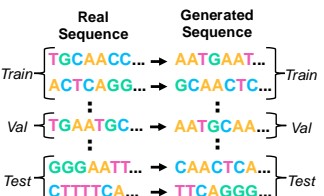

Figure 4: Visualization of data construction for context adherence evaluation.

**Data Construction.** We construct 5 datasets for this task. Each dataset contains 1000 non-overlapping genome sequences with 2000 base pairs from 10 unique species. As illustrated in Figure 4, we generate one sequence from each real sequence and use the same split to split the real and generated sequence into train, validation, and test sets with a ratio of $5 : 2 : 3$. To evaluate the generalizability of the models, among

Table 1: Species classification results of training on the real sequence and Val & Test on the generated sequences.

| | Judge | Real (Train) → Generated (Val & Test) | | | | Real → Real |
|---|---|---|---|---|---|---|
| | | GenomeOcean | GenSLM | Evo | Reorder | |
| *Unknown* | DNABERT-2 | **70.40 ± 4.68** | 35.59 ± 2.85 | 5.33 ± 0.47 | 27.40 ± 0.85 | 100.00 ± 0.00 |
| | HyenaDNA | **72.80 ± 1.30** | 43.34 ± 1.78 | 6.45 ± 1.76 | 45.62 ± 2.42 | 99.95 ± 0.11 |
| | NT-v2 | **75.08 ± 2.16** | 34.95 ± 3.82 | 5.55 ± 0.62 | 11.34 ± 0.88 | 99.89 ± 0.22 |
| | Caduceus | **52.63 ± 5.92** | 27.83 ± 2.61 | 5.15 ± 1.09 | 22.85 ± 4.04 | 100.00 ± 0.00 |
| *Known* | DNABERT-2 | **68.11 ± 1.56** | 34.39 ± 1.79 | 7.24 ± 1.49 | 17.85 ± 1.99 | 90.03 ± 0.59 |
| | HyenaDNA | **62.81 ± 0.95** | 35.47 ± 1.57 | 5.07 ± 0.75 | 29.85 ± 2.85 | 86.01 ± 0.72 |
| | NT-v2 | **66.63 ± 1.43** | 32.67 ± 3.66 | 5.61 ± 0.83 | 12.60 ± 1.75 | 88.61 ± 0.89 |
| | Caduceus | **62.87 ± 1.74** | 28.88 ± 0.96 | 7.02 ± 1.00 | 22.43 ± 2.90 | 81.30 ± 3.24 |

the 5 datasets, 2 contains all unknown or uncharacterized species, and the rest 3 contains all known species. We acquire all the genome sequences from the CAMI2 benchmark (Meyer et al., 2022).

We perform two complementary experiments using these datasets. In the first experiment, we train a model for species classification using the real sequences, while validating and testing on the generated sequences. This experiment evaluates how well the generated sequences retain the species-specific characteristics at the individual level. Specially, how many generated sequences retain the same species characteristics that the model discovered from the real sequences. In the complementary experiment, we examine the context adherence and characteristics retained at a distribution level. We instead train the species classification model using the generated sequences, while validating and testing on the real sequence. This experiment estimates how well we can classify the real sequences by aggregating the species-specific information. We also train, validate, and test each model on the real sequences to form the control group. Since sequences from the same species often have a similar composition (e.g., a similar ratio of A/T/C/G), in this task, we also use a simple baseline **Reorder**, which reorder all the characters of the input real sequence to produce a fake generated sequence.

One flaw of this evaluation method is that when the sequence generated by the model is highly close to or even copied from the given context, the model can achieve very good results. To rule out this possibility, we first visualize the closeness between the sequence generated by each model and the given context. As a comparison, we also calculate the closeness between the context and the real ground truth as a control group. Following previous works (Kang et al., 2015; Nissen et al., 2021), we measure the closeness of two sequences with the cosine similarity of their tetra-nucleotide frequency (TNF). TNF computes the frequency of each unique 4-mer as the sequence representation, so a generated

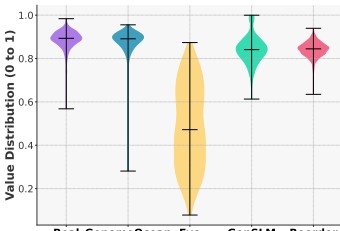

Figure 5: TNF similarity between context and generated sequence.

sequence that copies from the context will have a very high closeness with the context. Figure 5 shows the similarity distribution between 1000 pairs of context and generated sequences. As shown in the figure, GenomeOcean displays a similar pattern to the real data, except for a few outliers. We do not observe abnormally high closenesses of all the models from the figure. We take 4 discriminative genome foundation models as the judge of this task, including Nucleotide Transformers-V2 (Dalla-Torre et al., 2023), HyenaDNA (Nguyen et al., 2023), DNABERT-2 (Zhou et al., 2023), and Caduceus (Schiff et al., 2024). These models utilize different tokenization methods and model architecture so they may identify signals from diverse perspectives. We train each model on each set of real/generated sequences with 3 random seeds and report the mean and std of the 3 runs.

Table 1 shows the judges' macro F1 score when trained on the real sequences and tested on the generated sequence. The results of the complementary experiments are shown in Table 2. We aggregate the results on the 2 datasets with unknown species and 3 datasets with known species, respectively. Results on each dataset are presented in Section A.3. As shown in the table, based on the evaluation of all the judges, GenomeOcean generated sequences that contain better species-related information than the baselines. Over 60% of GenomeOcean-generated sequences can be correctly recognized with classifiers trained on the real sequences, showing its good capability in retaining species-relative information in each individual generation. When training the classifiers on the sequences generated by GenomeOcean, we achieve around 90% F1 scores compared to the ones

Table 2: Species classification results of training on the generated sequence and Val & Test on the real sequences.

| | Judge | Generated (Train) → Real (Val & Test) | | | | Real → Real |
|---|---|---|---|---|---|---|
| | | GenomeOcean | GenSLM | Evo | Reorder | |
| *Unknown* | DNABERT-2 | **95.71 ± 0.90** | 73.90 ± 1.75 | 10.97 ± 1.75 | 40.10 ± 4.85 | 100.00 ± 0.00 |
| | HyenaDNA | **90.05 ± 3.10** | 67.71 ± 3.97 | 5.08 ± 1.11 | 81.83 ± 3.26 | 99.95 ± 0.11 |
| | NT-v2 | **94.92 ± 1.71** | 71.64 ± 4.06 | 12.13 ± 1.65 | 54.80 ± 4.69 | 99.89 ± 0.22 |
| | Caduceus | **78.95 ± 4.70** | 47.08 ± 4.79 | 5.63 ± 2.43 | 55.64 ± 4.68 | 100.00 ± 0.00 |
| *Known* | DNABERT-2 | **81.46 ± 1.07** | 60.18 ± 2.49 | 12.32 ± 6.13 | 24.13 ± 5.03 | 90.03 ± 0.59 |
| | HyenaDNA | **73.73 ± 1.84** | 43.61 ± 3.67 | 7.05 ± 1.73 | 35.04 ± 1.97 | 86.01 ± 0.72 |
| | NT-v2 | **79.13 ± 2.03** | 57.11 ± 1.35 | 18.22 ± 2.69 | 24.08 ± 3.40 | 88.61 ± 0.89 |
| | Caduceus | **64.22 ± 5.63** | 38.07 ± 4.70 | 5.93 ± 2.28 | 37.96 ± 2.76 | 81.30 ± 3.24 |

trained on real data, suggesting that sequences generated by GenomeOcean effectively preserve species information at the distribution level. GenomeOcean consistently achieves much better performance than baselines. Moreover, we analyze the impact of context length on GenomeOcean to retrain species information in the generated sequences. Figure 6 shows the models' performance on the sequences generated from context length ranging from 500 to 16000 base pairs. As shown in the figure, GenomeOcean generates sequences with better species awareness as the length of context sequence increases, indicating its capability to understand and utilize long context.



Figure 6: Impact of context sequence lengths on the species information retaining in GenomeOcean generation. We use context sequences ranging from 500 base pairs to 16000 base pairs.

## 4.2 SIMILARITY TO GROUND-TRUTH

In this section, we evaluate the similarity between the sequence generated from an input context and the real sequence following the context (*i.e., ground-truth*). We measure the similarity by comparing the biological properties of the generated sequences and the ground-truth. We consider two properties of the sequences: length of the longest open reading frame (ORF) and codon usage bias. We present experiments on these two properties in Section 4.2.1 and Section 4.2.2, respectively.

### 4.2.1 OPEN READING FRAMES

Open Reading Frames, or ORFs, are important features within genetic sequences that play a crucial role in the process of protein production. In DNA sequence, each group of three nucleotide bases (e.g., ATG) is called a codon. An ORF is a continuous stretch of codons that starts with a "start" signal and ends with a "stop" signal. In between these signals, the ORF contains the instructions for building a specific protein. The length of an ORF is important because it directly relates to the size of the protein that could potentially be produced. Longer ORFs generally correspond to larger proteins, while shorter ORFs may result in smaller proteins or might not be used to make proteins at all. Genomics researchers often pay special attention to the longest ORF in a given sequence, as this can provide clues about whether the sequence is likely to be used to make proteins (coding) or serve other purposes in the cell (non-coding).

In sum, the length of the longest open reading frame serves as an indicator of a DNA sequence's functionality. An ideal generative model should produce sequences that demonstrate aligned characteristics in functionalities. Notably, the length of the longest ORFs often differs between coding and non-coding regions. To examine whether the models can generate sequences that adhere to the appropriate distribution, we construct two distinct datasets: one containing sequences from coding regions and the other from non-coding regions.

**Data Construction.** Consistent with our previous approach, we use a context sequence of 2000-bp and generate a corresponding 2000-bp sequence from each context. For the non-coding dataset, we source all available non-coding RNAs from the NONCODE database (Zhao et al., 2016). We apply a

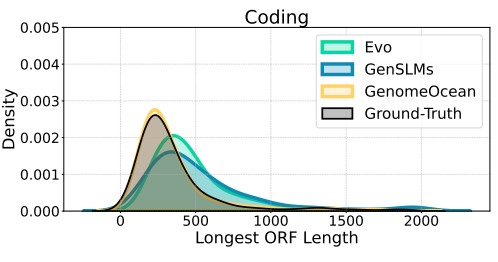
(a) Distribution on coding sequences.

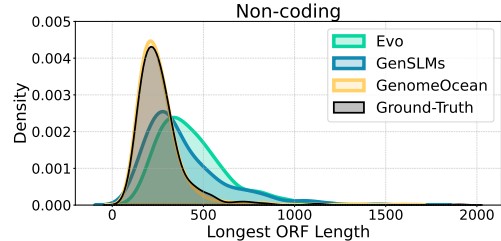
(b) Distribution on non-coding sequences.

Figure 7: Distribution of the longest ORF length on real and model-generated sequences.

filter to select sequences longer than 4000-bp and randomly choose 500 of them. Each RNA sequence is then converted to a DNA sequence by substituting *U* with *T*. We divide each 4000-bp sequence into two 2000-bp fragments, designating the first as the context and the second as the ground truth. For the coding dataset, we download 100 genomes from GenBank (Benson et al., 2012) and randomly select 500 4000-bp sequences containing ORFs. We apply the same sequence-splitting process to these coding sequences, resulting in 500 pairs of context and ground-truth sequences. These datasets allow us to estimate the models' ability to generate functionally appropriate sequences in varied contexts.

Figure 7 shows the distribution of the longest ORF length in the coding (7a) and non-coding (7b) datasets. We compare the distribution of the generated sequences from GenomeOcean, GenSLMs and Evo with the distribution of the ground-truth. As shown in the figure, ground-truth sequences in coding and non-coding datasets demonstrate distinct distribution of the longest ORF length. Compared to the coding dataset, shorter ORFs that range from 0 to 500-bp are much more frequent in the non-coding dataset. GenomeOcean accurately captures the distribution difference in these two dataset. The

Table 3: Pearson correlation of the generated ones and ground-truth on the longest ORF length.

| Model | Corr. ↑ |
|---|---|
| **GenomeOcean** | **12.18** |
| Evo | 7.51 |
| GenSLMs | 6.78 |

distribution of sequences generated by GenomeOcean shows a better alignment to the real distribution compared to the ones generated by Evo and GenSLMs. In both datasets, Evo and GenSLMs tends to generate sequences with longer ORFs than the real data. This observation demonstrate that GenomeOcean is able to understand the underlying function-related patterns in the context sequence. Moreover, we evaluate the sequences at the individual-level by computing the Pearson Correlation between the generated sequence and ground-truth. Table 3 shows the correlation scores. The sequences generated by all the models show a positive correlation with the real ones. Among the models, GenomeOcean achieves the best correlation. Yet the correlation is not significant, possibly due to the large difficulty of predicting the exact longest ORF lengths in the ground-truth, considering that there could be more than one *ground-truth* sequence given the same context due to the huge genomics diversity. In sum, genome foundation models like GenomeOcean can understand the underlying distribution of the context and produce biologically reasonable sequences at the distribution level. Yet, they are not able to consistently generate sequences that align with the ground-truth for each input context.

### 4.2.2 CODON USAGE BIAS

Codon usage bias refers to the phenomenon where certain codons (triplets of nucleotides that code for amino acids) are used more frequently than others, even though multiple codons can code for the same amino acid. This bias exists because different organisms, and even different genes within the same organism, have preferences for particular codons. Comparing the generated sequences to real genomes on codon usage bias helps us understand whether the generated sequence mirrors the natural patterns of codon choice. It estimates whether the generated sequence would function similarly to a real one. We quantify the codon usage bias of a sequence with a widely used method called Codon Adaptation Index (CAI) (Sharp & Li, 1987).

**Data Construction.** We randomly select 6 well-characterized microbial species for this evaluation. We compile 1 dataset with each species. For each species, we download all its strains that were published in 2024 from NCBI[2] to ensure they are not covered in the training data of all the models.

---

[2]https://www.ncbi.nlm.nih.gov/

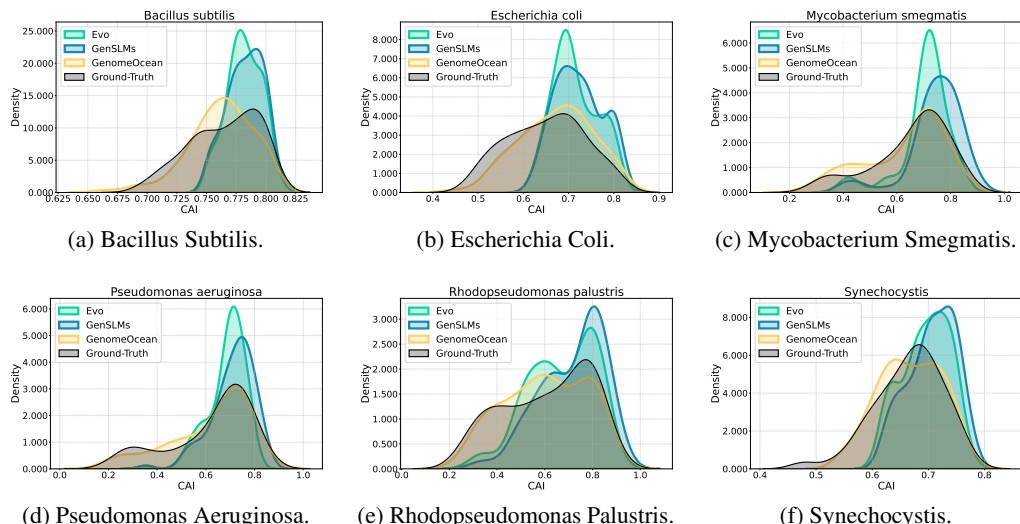

Figure 8: Distribution of codon usage bias measured by codon adaptation index (CAI). We compare all the models with the ground-truth on 6 different datasets.

We then randomly select 100 ORFs longer than 4200-bp from each species. The first 2100-bp of each ORF is used as the prompt, and the following 2100-bp is used as the ground truth. This data construction ensures the entire prompt and ground truth are inside an ORF, allowing a more accurate estimation of codon usage bias.

Figure 8 shows the models' results on 6 different species. Though different species has distinct codon usage bias as shown in the figure, the models are able to recognize the patterns from the context and generate sequences with similar bias. Among the models, sequences generated by GenomeOcean demonstrate codon usage bias patterns that are more similar to the ground truth, showing it capability of understanding the context and produce corresponding sequences. Furthermore, Table 4 shows the correlation between codon usage bias of the generated and ground-truth sequences. GenSLMs obtain a

Table 4: Pearson correlation of the generated ones and ground-truth on codon usage bias.

| Model | Corr. ↑ |
|---|---|
| **GenomeOcean** | **32.06** |
| **Evo** | 14.18 |
| **GenSLMs** | -5.83 |

negative correlation coefficient, showing that it may not consider codon usage biased when generating sequences. Aligning with our previous observation, GenomeOcean demonstrates a good capability to generate sequences with appropriated codon usage bias.

## 5 CONCLUSION

We present GenomeOcean, an open and efficient generative genome foundation model capable of producing context-adherent genome sequences. Through a suite of automated experiments, we demonstrate its ability to discern the underlying distribution of context sequences and generate sequences that retain species-specific characteristics, contain appropriate open reading frames, and incorporate desired codon usage bias. Our efficiency-oriented design, encompassing tokenization, model architecture, and inference framework, enables GenomeOcean to generate over 12,000 base pairs per second on a single NVIDIA A100 80GB GPU. This represents an approximately 80× increase in inference throughput compared to existing models of similar size. The combination of highly optimized efficiency and high-quality genome generation opens up new possibilities for previously infeasible research involving genome generation.

**Limitation and future works.** Our evaluation primarily relies on automated experiments utilizing existing genome foundation models and quantitative metrics. While these experiments demonstrate GenomeOcean's advantages over existing models, they do not fully assess its efficacy in real-world applications. More rigorous and comprehensive evaluations, in collaboration with biologists, are essential to assess the model's performance in fine-grained genome understanding and synthesis of novel sequences with desired traits. As a manuscript targeting the machine learning community, we leave these in-depth biological evaluations to future studies.

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

# A APPENDIX

## A.1 MODEL ARCHITECTURE IN PRELIMINARY EXPERIMENTS

Table 5 shows the hyperparameters of the Mamba, Transformer, and MoE Transformers model we used in our preliminary experiments. We create the Mamba and Transformer in this way to allow similar training costs. We also make the Transformer and MoE Transformer have the same hidden size and number of layers.

Table 5: Hyperparameters of Models in Preliminary Experiments.

| Model Size | Mamba | Transformer | MoE Transformer |
|---|---|---|---|
| hidden size | 768 | 768 | 768 |
| intermediate size | 1536 | 3072 | 3072 |
| num. attention heads | N/A | 12 | 12 |
| num. query and key heads | N/A | 6 | 6 |
| num. layers | 24 | 12 | 22 |
| num. experts | N/A | N/A | 8 |
| num. experts activate | N/A | N/A | 2 |
| num. parameters | 93M | 112M | 702M |

## A.2 RESULTS OF DIFFERENT TOKENIZATION METHODS

In this section, we present the detailed results of each tokenizer as the DNA feature extractor. We use the token frequency as the DNA feature and train a multi-layer perceptron (MLP) as the classifier. This evaluation examines each tokenizer's expressiveness. We use the F1 score and Matthews correlation coefficient as the measures for different datasets, following (Zhou et al., 2023). We summarize the results with the used measure in Table 6.

## A.3 DETAILED RESULTS ON SPECIES CLASSIFICATION

In this section, we present the detailed species classification results of all judges in each dataset. We run each experiments with 3 random seeds and report the mean and std of the 3 runs. We summarize the results in Table 7 and 8. We observe a consistent pattern on all 5 datasets with distinct species.

## A.4 IMPACT OF CONTEXT SEQUENCE LENGTH ON SPECIES CLASSIFICATION

In this section, we present the models' performance on the problem of species classification. To understand the impact of context length on GenomeOcean in generating species-aware sequences, for each dataset, we respectively use the last 500, 1000, 2000, 4000, 8000, and 16000 base pairs of each context sequence as the input for model generation. We present the model's results on the 3 datasets with known species, as the sequence lengths are not long enough for this experiment in the datasets with unknown species. We also aggregate the results on the three datasets in Table 9 and 10. We observe consistent performance improvement as the context sequence becomes longer, indicating GenomeOcean's capability to understand and utilize longer contexts.

Table 6: Performance of different tokenization methods on GUE benchmark. Here is the list of abbreviations: (1) Char.-level: Character-level; (2) Overlap. 6-mer: Overlapping 6-mer; (3) Non. 6-mer: Non-overlapping 6-mer; (4) Non. 3-mer: Non-overlapping 3-mer; (5) BPE: Byte-pair Encoding; (6) E.M.P.: Epigenetic Marks Prediction; (7) P.D.: Promoter Detection; (8) C.P.D.: Core Promoter Detection; (9) T.F.P. (H.): Transcription Factor Prediction (Human); (10) T.F.P. (M.): Transcription Factor Prediction (Mouse); (11) F1: F1 score; (12) MCC: Matthews correlation coefficient.

| Dataset | | Tokenization | | | | |
| --- | --- | --- | --- | --- | --- | --- |
| | | Char.-level | Overlap. 6-mer | Non. 6-mer | Non. 3-mer | BPE |
| E.M.P. (MCC) | H3 | 52.87 | 69.74 | 52.84 | 62.45 | 60.04 |
| | H3K14ac | 7.14 | 50.80 | 27.63 | 35.12 | 42.92 |
| | H3K36me3 | 10.90 | 49.76 | 34.94 | 41.19 | 46.89 |
| | H3K4me1 | 15.54 | 38.61 | 22.64 | 29.02 | 37.04 |
| | H3K4me2 | 27.12 | 36.79 | 19.92 | 22.64 | 35.77 |
| | H3K4me3 | 7.98 | 41.19 | 18.61 | 20.39 | 40.49 |
| | H3K79me3 | 24.33 | 61.79 | 44.08 | 50.81 | 55.39 |
| | H3K9ac | 28.89 | 51.42 | 31.06 | 39.08 | 47.01 |
| | H4 | 39.68 | 71.88 | 55.96 | 69.69 | 61.49 |
| | H4ac | 17.62 | 48.40 | 27.57 | 32.84 | 44.98 |
| P.D. (MCC) | all | 50.66 | 74.71 | 28.68 | 29.59 | 21.33 |
| | notata | 57.68 | 85.63 | 77.67 | 78.26 | 77.83 |
| | tata | 30.31 | 38.01 | 68.24 | 69.04 | 67.81 |
| C.P.D. (MCC) | all | 42.98 | 54.34 | 45.82 | 50.99 | 45.90 |
| | notata | 47.62 | 57.61 | 49.45 | 55.18 | 50.37 |
| | tata | 38.40 | 39.59 | 50.88 | 40.34 | 39.59 |
| T.F.P. (H.) (MCC) | 0 | 11.69 | 58.60 | 44.84 | 49.85 | 50.78 |
| | 1 | 13.70 | 61.53 | 49.72 | 52.93 | 53.14 |
| | 2 | 9.38 | 58.93 | 39.17 | 37.00 | 41.22 |
| | 3 | 12.03 | 41.94 | 29.18 | 37.00 | 32.46 |
| | 4 | 18.74 | 66.30 | 47.14 | 46.15 | 49.13 |
| T.F.P. (M.) (MCC) | 0 | -4.18 | 50.18 | 22.66 | 14.01 | 28.48 |
| | 1 | 0.00 | 71.31 | 53.92 | 41.18 | 60.89 |
| | 2 | -9.11 | 71.85 | 50.13 | 35.77 | 52.34 |
| | 3 | -8.39 | 63.24 | 33.63 | 13.93 | 45.64 |
| | 4 | -2.34 | 36.02 | 16.62 | 12.75 | 17.97 |
| Virus (F1) | Covid | 14.27 | 65.90 | 51.96 | 44.73 | 68.52 |
| Splice (F1) | Reconstruct | 2.93 | 35.52 | 32.15 | 20.56 | 30.16 |
| Mean | | 19.94 | 55.41 | 40.25 | 39.92 | 46.46 |

Table 7: Species classification results of training on the generated sequence and Val & Test on the real sequences. Datasets 1 and 2 contain all unknown or uncharacterized species, and the other 3 datasets contain all known species. The measure is the F1 score.

| Dataset | Judge | Generated (Train) → Real (Val & Test) | | | | Real → Real |
|---|---|---|---|---|---|---|
| | | GenomeOcean | GenSLM | Evo | Reorder | |
| 1 | DNABERT-2 | **94.40 ± 1.04** | 67.52 ± 2.40 | 8.25 ± 2.29 | 41.47 ± 5.89 | 100.00 ± 0.00 |
| | HyenaDNA | **85.76 ± 4.32** | 68.72 ± 5.47 | 4.63 ± 0.89 | 75.84 ± 3.66 | 100.00 ± 0.00 |
| | NT-v2 | **91.96 ± 2.22** | 69.33 ± 3.91 | 11.73 ± 0.52 | 45.63 ± 3.32 | 99.78 ± 0.31 |
| | Caduceus | **77.49 ± 5.13** | 45.64 ± 2.71 | 3.64 ± 1.35 | 60.28 ± 6.43 | 100.00 ± 0.00 |
| 2 | DNABERT-2 | **97.02 ± 0.73** | 80.28 ± 0.61 | 13.69 ± 0.95 | 54.72 ± 3.52 | 100.00 ± 0.00 |
| | HyenaDNA | **98.33 ± 0.72** | 66.70 ± 1.24 | 5.53 ± 1.29 | 87.82 ± 2.80 | 99.89 ± 0.16 |
| | NT-v2 | **97.88 ± 0.96** | 73.95 ± 4.20 | 12.52 ± 2.27 | 63.97 ± 5.75 | 100.00 ± 0.00 |
| | Caduceus | **80.41 ± 4.22** | 48.52 ± 6.21 | 7.61 ± 3.16 | 50.99 ± 1.59 | 100.00 ± 0.00 |
| 3 | DNABERT-2 | **90.86 ± 0.53** | 64.59 ± 3.76 | 14.00 ± 8.62 | 24.47 ± 5.64 | 96.21 ± 0.84 |
| | HyenaDNA | **82.26 ± 2.82** | 43.52 ± 5.88 | 5.25 ± 1.41 | 34.12 ± 0.76 | 91.04 ± 1.02 |
| | NT-v2 | **89.30 ± 3.10** | 66.34 ± 1.19 | 17.92 ± 2.90 | 25.57 ± 0.15 | 95.57 ± 0.78 |
| | Caduceus | **60.94 ± 8.61** | 40.36 ± 5.78 | 5.11 ± 2.36 | 36.18 ± 1.14 | 90.55 ± 0.44 |
| 4 | DNABERT-2 | **61.87 ± 0.40** | 46.94 ± 1.92 | 13.27 ± 2.57 | 18.59 ± 1.32 | 78.40 ± 0.41 |
| | HyenaDNA | **50.06 ± 0.45** | 30.58 ± 2.31 | 7.43 ± 0.29 | 34.40 ± 1.09 | 71.44 ± 0.40 |
| | NT-v2 | **57.36 ± 1.66** | 41.05 ± 0.57 | 14.94 ± 0.45 | 22.03 ± 4.91 | 74.72 ± 0.85 |
| | Caduceus | **50.80 ± 1.63** | 27.09 ± 0.52 | 6.38 ± 2.10 | 37.93 ± 4.09 | 60.63 ± 5.53 |
| 5 | DNABERT-2 | **91.66 ± 1.73** | 69.00 ± 0.91 | 9.70 ± 5.63 | 29.33 ± 6.51 | 95.48 ± 0.41 |
| | HyenaDNA | **88.86 ± 1.40** | 56.73 ± 0.66 | 8.46 ± 2.63 | 36.60 ± 3.15 | 95.54 ± 0.61 |
| | NT-v2 | **90.73 ± 0.15** | 63.94 ± 1.93 | 21.80 ± 3.62 | 24.65 ± 3.24 | 95.54 ± 1.03 |
| | Caduceus | **80.92 ± 4.26** | 46.76 ± 5.72 | 6.29 ± 2.38 | 39.76 ± 2.20 | 92.72 ± 0.84 |

Table 8: Species classification results of training on the real sequences and Val & Test on the generated sequence. Datasets 1 and 2 contain all unknown or uncharacterized species, and the other 3 datasets contain all known species. The measure is the F1 score.

| Dataset | Judge | Real (Train) → Generated (Val & Test) | | | | Real → Real |
|---|---|---|---|---|---|---|
| | | GenomeOcean | GenSLM | Evo | Reorder | |
| 1 | DNABERT-2 | **75.75 ± 0.15** | 37.44 ± 3.67 | 4.65 ± 0.38 | 31.11 ± 1.09 | 100.00 ± 0.00 |
| | HyenaDNA | **76.86 ± 0.67** | 41.60 ± 0.92 | 6.76 ± 1.54 | 41.44 ± 1.40 | 100.00 ± 0.00 |
| | NT-v2 | **74.45 ± 1.66** | 34.96 ± 3.82 | 5.55 ± 0.62 | 11.34 ± 0.88 | 99.78 ± 0.31 |
| | Caduceus | **55.85 ± 5.98** | 27.39 ± 2.22 | 5.39 ± 0.98 | 25.39 ± 4.80 | 100.00 ± 0.00 |
| 2 | DNABERT-2 | **65.04 ± 6.61** | 33.74 ± 1.68 | 6.00 ± 0.54 | 23.69 ± 0.52 | 100.00 ± 0.00 |
| | HyenaDNA | **68.73 ± 1.71** | 45.07 ± 2.34 | 6.13 ± 1.95 | 49.80 ± 3.13 | 99.89 ± 0.16 |
| | NT-v2 | **75.71 ± 2.56** | 34.94 ± 3.82 | 5.55 ± 0.62 | 11.34 ± 0.88 | 100.00 ± 0.00 |
| | Caduceus | **49.41 ± 5.85** | 28.27 ± 2.95 | 4.90 ± 1.19 | 20.31 ± 3.10 | 100.00 ± 0.00 |
| 3 | DNABERT-2 | **76.56 ± 0.53** | 35.55 ± 2.35 | 6.14 ± 1.42 | 20.12 ± 1.86 | 96.21 ± 0.84 |
| | HyenaDNA | **71.90 ± 0.25** | 38.86 ± 1.48 | 5.52 ± 0.64 | 22.44 ± 2.80 | 91.04 ± 1.02 |
| | NT-v2 | **73.70 ± 1.96** | 33.83 ± 3.51 | 6.39 ± 0.65 | 14.37 ± 1.44 | 95.57 ± 0.78 |
| | Caduceus | **71.53 ± 2.48** | 30.81 ± 0.27 | 7.73 ± 0.73 | 21.34 ± 3.49 | 90.55 ± 0.44 |
| 4 | DNABERT-2 | **52.98 ± 1.29** | 31.30 ± 1.80 | 6.97 ± 0.81 | 17.59 ± 2.34 | 78.40 ± 0.41 |
| | HyenaDNA | **44.63 ± 1.26** | 28.27 ± 1.45 | 4.55 ± 1.07 | 30.50 ± 2.56 | 71.44 ± 0.40 |
| | NT-v2 | **50.88 ± 0.19** | 25.52 ± 1.82 | 4.69 ± 1.21 | 13.48 ± 2.61 | 74.72 ± 0.85 |
| | Caduceus | **44.28 ± 1.52** | 21.25 ± 1.04 | 4.27 ± 1.22 | 25.54 ± 0.39 | 60.63 ± 5.53 |
| 5 | DNABERT-2 | **74.80 ± 2.32** | 36.33 ± 0.93 | 8.60 ± 2.00 | 15.84 ± 1.71 | 95.48 ± 0.41 |
| | HyenaDNA | **71.91 ± 1.03** | 39.28 ± 1.76 | 5.13 ± 0.36 | 36.60 ± 3.15 | 95.54 ± 0.61 |
| | NT-v2 | **75.30 ± 1.51** | 38.66 ± 4.96 | 5.76 ± 0.40 | 9.95 ± 0.57 | 95.54 ± 1.03 |
| | Caduceus | **72.81 ± 0.77** | 34.57 ± 1.28 | 9.07 ± 1.00 | 20.41 ± 3.58 | 92.72 ± 0.84 |

Table 9: Impact of Context Length: We train on the generated sequence by GenomeOcean and Val & Test on the real sequences with the species classification task. We use the context length of 500, 1000, 2000, 4000, 8000, and 16000. The measure is the F1 score.

| Dataset | Judge | Generated (Train) → Real (Val & Test) | | | | | |
|---|---|---|---|---|---|---|---|
| | | 500 | 1000 | 2000 | 4000 | 8000 | 16000 |
| 3 | DNABERT-2 | 80.68 ± 0.19 | 91.18 ± 0.82 | 90.86 ± 0.53 | 90.39 ± 0.50 | 91.21 ± 1.61 | 92.98 ± 1.49 |
| | HyenaDNA | 62.91 ± 4.31 | 80.45 ± 1.00 | 82.26 ± 2.82 | 83.60 ± 1.06 | 87.35 ± 0.63 | 86.96 ± 0.49 |
| | NT-v2 | 78.11 ± 3.02 | 88.27 ± 2.27 | 89.30 ± 3.10 | 90.24 ± 0.83 | 90.39 ± 0.34 | 91.69 ± 0.41 |
| | Caduceus | 42.26 ± 7.80 | 61.48 ± 8.37 | 60.94 ± 8.61 | 73.90 ± 1.48 | 70.45 ± 4.54 | 76.80 ± 2.51 |
| 4 | DNABERT-2 | 57.06 ± 0.67 | 59.38 ± 2.20 | 61.87 ± 0.40 | 62.08 ± 2.11 | 64.26 ± 2.79 | 66.90 ± 1.76 |
| | HyenaDNA | 46.53 ± 5.13 | 51.11 ± 0.76 | 50.06 ± 0.45 | 52.17 ± 2.37 | 54.70 ± 1.55 | 54.90 ± 2.43 |
| | NT-v2 | 49.72 ± 1.95 | 57.37 ± 3.59 | 57.36 ± 1.66 | 56.46 ± 1.38 | 59.04 ± 3.81 | 61.39 ± 0.33 |
| | Caduceus | 42.66 ± 2.11 | 46.00 ± 2.11 | 50.80 ± 0.16 | 46.62 ± 2.95 | 46.82 ± 3.04 | 48.03 ± 2.96 |
| 5 | DNABERT-2 | 82.40 ± 0.83 | 91.02 ± 0.46 | 91.66 ± 1.73 | 92.63 ± 0.56 | 92.52 ± 0.11 | 90.19 ± 0.79 |
| | HyenaDNA | 61.76 ± 6.20 | 80.71 ± 3.23 | 88.86 ± 1.40 | 91.47 ± 0.39 | 91.05 ± 1.25 | 91.34 ± 1.03 |
| | NT-v2 | 78.06 ± 1.45 | 85.12 ± 2.06 | 90.73 ± 0.15 | 88.74 ± 2.34 | 89.51 ± 0.74 | 89.25 ± 2.17 |
| | Caduceus | 49.47 ± 5.08 | 63.79 ± 6.03 | 80.92 ± 4.26 | 77.17 ± 1.22 | 85.55 ± 3.36 | 88.90 ± 1.29 |

Table 10: Impact of Context Length: We train on the real sequences and Val & Test on the generated sequence by GenomeOcean with the species classification task. We use the context length of 500, 1000, 2000, 4000, 8000, and 16000. The measure is the F1 score.

| Dataset | Judge | Real (Train) → Generated (Val & Test) | | | | | |
|---|---|---|---|---|---|---|---|
| | | 500 | 1000 | 2000 | 4000 | 8000 | 16000 |
| 3 | DNABERT-2 | 56.46 ± 0.75 | 68.45 ± 2.76 | 76.56 ± 0.53 | 79.31 ± 2.66 | 82.43 ± 0.43 | 84.08 ± 1.08 |
| | HyenaDNA | 49.44 ± 1.96 | 63.92 ± 0.37 | 71.90 ± 0.25 | 76.18 ± 0.45 | 80.16 ± 0.88 | 77.73 ± 1.07 |
| | NT-v2 | 50.44 ± 1.98 | 63.97 ± 2.46 | 73.70 ± 1.96 | 80.33 ± 0.48 | 81.57 ± 1.67 | 81.09 ± 2.62 |
| | Caduceus | 48.55 ± 2.42 | 59.36 ± 1.69 | 71.53 ± 2.48 | 75.32 ± 4.22 | 71.26 ± 2.41 | 72.31 ± 2.34 |
| 4 | DNABERT-2 | 40.87 ± 0.32 | 49.76 ± 2.01 | 52.98 ± 1.29 | 52.31 ± 1.77 | 53.75 ± 3.27 | 53.73 ± 0.42 |
| | HyenaDNA | 38.54 ± 1.23 | 42.98 ± 3.70 | 44.63 ± 1.26 | 48.69 ± 1.25 | 51.24 ± 1.88 | 50.65 ± 1.50 |
| | NT-v2 | 39.63 ± 3.12 | 45.35 ± 2.19 | 50.88 ± 0.19 | 52.90 ± 3.76 | 54.04 ± 5.14 | 55.64 ± 2.13 |
| | Caduceus | 34.77 ± 1.47 | 43.13 ± 2.27 | 44.28 ± 1.52 | 45.80 ± 1.75 | 50.69 ± 2.67 | 47.74 ± 2.52 |
| 5 | DNABERT-2 | 55.22 ± 2.89 | 64.87 ± 2.33 | 74.80 ± 2.32 | 79.71 ± 0.89 | 83.49 ± 1.21 | 79.68 ± 1.15 |
| | HyenaDNA | 50.02 ± 3.34 | 60.94 ± 0.65 | 71.91 ± 1.03 | 73.82 ± 1.25 | 81.89 ± 1.24 | 77.61 ± 2.05 |
| | NT-v2 | 56.36 ± 1.05 | 64.98 ± 1.50 | 75.30 ± 1.51 | 74.50 ± 0.75 | 84.53 ± 1.52 | 82.79 ± 1.22 |
| | Caduceus | 51.36 ± 1.80 | 60.77 ± 0.93 | 72.81 ± 0.77 | 71.30 ± 1.04 | 76.25 ± 4.61 | 74.09 ± 1.00 |