# OpenReview forum: "GenomeOcean: Efficient Foundation Model for Genome Generation"
_ICLR.cc/2025/Conference — Submitted to ICLR 2025_

### Official Review · Reviewer_TeSQ · 2024-10-18

**Soundness:** 1
**Presentation:** 2
**Contribution:** 3
**Rating:** 3
**Confidence:** 4

**Summary:**

The authors present GenomeOcean, a genomic foundation model (FM) that is trained on genomic sequences from environmental samples. GenomeOcean is built upon the transformer decoder architecture and integrates several recent LLM techniques to improve training and inference speed. The authors benchmark several design decisions, such as the tokenizer and training objective, on downstream performance on the Genome Understanding Evaluation benchmark. After pre-training, the authors tackle the problem of biologically plausible sequence generation. Highlighting the lack of a ground truth without biological experimentation, the authors propose two metrics by which to evaluate the quality of generated sequences: the ability for existing discriminative models to classify generated sequences based on species, and distributional similarity of generated sequences to real sequences in opening reading frame length and codon bias.

**Strengths:**

- The authors benchmark several architectural decisions that are commonly made in genomic foundation modelling. Despite the concerns highlighted below, this type of benchmark is likely of high significance to the community once concerns about the downstream tasks are addressed.
- The authors successfully combine several recent efficiency improvements to vastly improve sequence generation speed. This capability could be relevant for real world applications in synthetic biology or large in-silico experiments.
- Proposed metrics for evaluating quality of generated sequences are to my knowledge novel and a good initial attempt at addressing a difficult yet relevant problem. GenomeOcean shows outperformance in the selected metrics compared to existing generative genomic foundation models.
- The addition of the Reorder baseline alleviates concerns about shortcut learning on chemical characteristics (e.g. GC content) during generated sequence quality evaluation.

**Weaknesses:**

As the paper's contribution lies in its application to biology and genomics rather than novel methodological advancements, the biological accuracy of the experiments and generated sequences are of high significance to the overall impact of the paper. Several experimental concerns are highlighted below:

- The paper omits any statistical or textual description of the pre-training dataset, other than the method of collection, making it difficult to form intuition about the biological validity of experiments and judge the significance of the method. It is unknown how much of this pre-training dataset (and consequently GenomeOcean) overlaps with existing genome foundation models. As the authors note, the dataset will be released and will likely be an extremely valuable resource for the community, but as it stands it is difficult to assess the paper's contributions in this aspect.

- While the evaluation of common genomic FM modelling decisions is a valuable contribution to the community, the choice to evaluate performance on the Genome Understanding Evaluation benchmark may dilute the insights gained. Based on my understanding of the benchmark from the original paper, the train/val/test splits in the datasets are not made while accounting for homology or sequence similarity. This filtering step is common in other biological domains such as pLMs. Without this step, data leakage may occur, and it becomes unclear whether the models are truly capturing biological function / structure. Consequently, it is unclear whether the modelling decisions in the preliminary experiments contribute towards better representations.

- Although the proposed evaluation metric in Section 4.1 is novel to my knowledge (though perhaps it could be formalized into something resembling the inception score or FID) and interesting from a methodology perspective, using other genomic foundation models as a method to discriminate generated sequences may be a limitation, as the metric would suffer from inherent biases in the existing FMs. This may detract from the metric, given the discourse in community on the overall validity of genomic foundation models. A simple baseline agnostic of existing FMs such as phylogenetic distance between sequences would be interesting to see.

- While the proposed evaluation metrics have a reasonable setup, the tasks evaluated (e.g. species classification) lack strong correlation with the functional validity of the generated sequences, which is highly relevant for real world utility of GenomeOcean in my opinion.

- Minor, but claims in the introduction that genomic foundation models outperform classic supervised models are unsubstantiated. For splice site prediction specifically, **none of the in-line citations provided mention splice site detection.** Moreover, to my knowledge, there has been no comparison of genomic foundation models against more modern splice site predictors, nor on the more realistic splice site prediction task described in SpliceAI / Pangolin. As such, the overall claim may be pre-mature.

**Questions:**

- I am wondering if the BPE vs character level encoding experiment in Section 2.1 properly controls for pre-training context window. Since it was not clear from the text, I am wondering whether the character-level and BPE tokenizers observe the same number of DNA nucleotides during pre-training, or if they observe the same number of input tokens. Due to the high compression factor of BPE, the latter experimental design would allow the BPE model to access a much longer context. If this were the case, we would be assessing the benefits of BPE in terms of computational efficiency rather than any true representational gains as the paper may imply.
- It is surprising that the Evo model is outperformed by the Reorder baseline, and it would be interesting (but not necessary to the review) to hear if the authors have any insight into this phenomena based on their experiments.
- It is unclear exactly how the genomic sequences using in Section 4.1 were selected. Were these 2000bp sequences taken from coding regions, or were they taken from randomly sampled positions in the genomes from the dataset? The improvement in performance as a function of sequence length is also unexpected, as one might expect the quality of generated sequence to decrease as a function of length, and I am wondering if the authors could provide intuition on this aspect.

Overall, the paper tackles important and relevant problems in genomic modelling, and I believe the contributions are sufficient for eventual publication given several experimental concerns are addressed or rebutted.

---

### Official Review · Reviewer_LKnb · 2024-10-22

**Soundness:** 2
**Presentation:** 3
**Contribution:** 2
**Rating:** 3
**Confidence:** 4

**Summary:**

GenomeOcean is a generative model for genomes. Unlike many genome sequence models, this model is trained on a diverse set of genomes sampled from different ecosystems.

**Strengths:**

- GenomeOcean appears to have a large advantage over other generative models in creating sequences
- GenomeOcean can generate genome sequences quickly, and is 80x faster than existing models of similar size.

**Weaknesses:**

- Poorly motivated in terms of use cases (see my first question)
- Using GUE to determine which tokenization, model architecture, and training objective to be used for the generative model seems arbitrary and potentially incongruous. The representations learned using these modeling decisions will be optimized for the relatively narrow set of tasks presented in GUE, and may have very little generalizability.
- Given the prevalence of CNN-based models in genome sequence modeling, the authors should also consider CNN-based architectures such as GPN (Benegas et. al, 2023) or LOGO (Yang et. al, 2022).
- The results from the biological properties investigation are not significant
- As the authors correctly state, it is difficult to validate the biological plausibility of genome sequences. Thus, there needs to be a more comprehensive effort to evaluate the “goodness” of the generated genomes in this paper. Currently, there is too much emphasis on model design and not enough emphasis on validation.
- The main context benchmarks in 4.1 are inherently better suited to the biases of GenomeOcean, which has likely seen a larger diversity of sequences across domains, whereas  Evo and GenSLM are only trained on prokaryotes and viruses, respectively.


Benegas, G., Batra, S. S., & Song, Y. S. (2023). DNA language models are powerful predictors of genome-wide variant effects. Proceedings of the National Academy of Sciences, 120(44), e2311219120.

Yang, M., Huang, L., Huang, H., Tang, H., Zhang, N., Yang, H., Wu, J., and Mu, F. (2022). Integrating convolution and self-attention improves language model of human genome for interpreting non-coding regions at base-resolution.Nucleic Acids Research 50, e81–e81.

**Questions:**

- My main issue with this paper is that it is weakly motivated - why are generated genome sequences from GenomeOcean useful? The other generative models discussed in the paper, such as GenSLM and Evo, are employed in highly specific and impactful contexts—tracking Covid evolution and synthetic molecule creation, respectively. These targeted applications highlight the value of their generative capabilities. Without demonstrating a similarly practical and realistic use case, the contributions of GenomeOcean remain quite limited.
- The details of “Reorder” are not mentioned. How does the reordering of characters work in the instance of this paper?
- Given that paper claims that reference genomes possess inherent biases, how does GenomeOcean perform when trained only on reference genomes? Providing evidence of a difference in performance for the model trained on reference genomes vs. the samples used in the paper would be a more compelling result.
- Is the correlation between the generated sequences and ground truth sequences significant on codon usage bias? What are the p-values for all of the correlations computed in the paper?
- What does it mean to be a “known” vs. “unknown” species in Table 2? What is the significance of this distinction in terms of the species classification task? Please provide further information on the samples that GenomeOcean is trained on

---

### Official Review · Reviewer_ok5b · 2024-11-03

**Soundness:** 2
**Presentation:** 2
**Contribution:** 2
**Rating:** 5
**Confidence:** 3

**Summary:**

First, the authors explore the factors that make DNA “foundation models” perform better by understanding the impact of tokenization and causal vs masked language modeling. Next, the authors optimize inference speed of a Transformer model and train it on a large dataset of environmental samples. They then evaluate generative performance on a number of biologically relevant tasks.

**Strengths:**

I think the observation of the relative performance of causal language model performance being better than masked language is very interesting (Figure 3a and 3b).

I think the species identification task in Table 1 is an interesting approach.

I think the codon usage experiment in Figure 8 is interesting. I do think it conveys that their model better adheres to context.

**Weaknesses:**

Figure 1 on it’s own is not a great selling point to me, as it does not tell me the quality of the results. random.choice([“A”,”T”,”C”,”G”]) with an associated single and bigram base frequency is most certainly faster and has fewer parameters!

The test described in Figure 2b is incongruent with the rest of the paper. Here the authors train a descrimitive model using different DNA representations as input. It’s obvious that the character level input will do poorly because there is no context window. In modern generative models, there’s a huge context window. It’s difficult to see how this discriminative test translates to the modern generative modeling case. Especially with what is shown in Figure 2c, why not just do that set of experiments for all different DNA tokenizers?

Since the speed of GenomeOcean is strongly advertised, it’d be great to have an ablation study of the improvements that made the model what % faster. All of these are listed in section 3.1. This is a relatively easy experiment to do, because you don’t have to train the model for this; it’s just inference speed.

For section 4.1 and Table 1, did you hold out those species or even representative genus while training the generative models?

I think the ORF length analysis is quite weak. The Ground Truth should be the dataset the model was trained on, so a comparison of the models can be done. Evo was trained on OpenGenome, which is just prokaryotic sequences. Did you filter in your “ground truth” dataset for this? Which did you use?

**Questions:**

“On the other hand, computational efficiency is essential. Generating novel, realistic, and biologically valid DNA sequences often requires extensive experimentation.” I don’t think the bottleneck is on generative speed, it is on biological validation, as you note.

What is happening in Figure 2c? Is this explained in the text? What is Number of Wins?

How do you interpret training loss in Figure 3c? Wouldn’t validation loss of some held out sequences make more sense?

Where is the training loss curve for DNABERT-2?

What is the average contig length of the environmental samples taken? Are they assembled genomes, or minimally large contigs? There’s no point of a long-context model if you aren’t training on long contigs. What is the distribution of Kingdoms? There’s probably not a lot of eukaryotic sequences, which is important for many engineering contexts.

I’m wondering if the species identification approach is a good proxy for designing sequences, as you’ve outlined in the intro on the use of these models. It doesn’t matter if the generated sequences adhere to the same species, they just need to be functional. In the case of a protein, you can easily make a transgenic organism that will easily express and fold some foreign protein, even if it isn’t codon optimized for that species.

“Following previous works (Kang et al., 2015; Nissen et al., 2021), we measure the closeness of two sequences with the cosine similarity of their tetranucleotide frequency (TNF).” Why not just try to align them? In this evaluation method, you still would not detect perfect or near matches. Just BLAST, no?

---

### Official Review · Reviewer_wSUg · 2024-11-03

**Soundness:** 3
**Presentation:** 4
**Contribution:** 1
**Rating:** 3
**Confidence:** 5

**Summary:**

In this work, the authors introduce GenomeOcean, a 4-billion parameter autoregressive transformer model trained to generate DNA sequences. Notably, the model is trained on a newly curated dataset of large-scale environmental samples collected from diverse ecosystems, including oceans, lakes, forests, and soils. The authors conduct a thorough ablation study to determine the optimal tokenizer, model architecture, and training objective. They emphasize GenomeOcean's ability to generate sequences in-context, meaning it can be prompted with an existing DNA sequence fragment, and highlight its efficient inference speed achieved through the utilization of high-performance libraries. Furthermore, the authors develop a benchmark comprising three tasks designed to assess the model's ability to (1) generate sequences representative of a given species, (2) reproduce the length distributions of Open Reading Frames (ORFs) found in nature, and (3) reproduce codon usage distributions observed in nature. GenomeOcean is compared against existing methods, Evo and GenSLMs, demonstrating improved performance across the benchmark.

**Strengths:**

The paper is well-written and well-structured, making it easy to read and understand. The figures are of high quality, and the overall presentation is excellent. I particularly appreciate the initial sections where the authors introduce the relevant literature, existing models, and their design choices regarding datasets, tokenization, model architecture, and training objectives. The ablation study is also well-executed and thoroughly detailed. I would highly recommend these sections to newcomers in the field as they provide an excellent and comprehensive tutorial on building DNA foundation/generative models. The technical decisions seem sound, and I trust the overall technical quality of the work.

**Weaknesses:**

However, the paper exhibits the following limitations:

- The paper lacks significant research contributions. The authors repeatedly claim high inference speed as a major contribution. While they demonstrate efficient code, they primarily utilize existing codebases. Consequently, this represents good engineering practice rather than a novel research contribution. A true contribution would involve proposing a new method for accelerating such models, specifically within the context of DNA generation. The second claimed contribution is the ablation study; however, similar ablations have been performed extensively in the literature, such as in NT, DNABert, HyenaDNA, and Evo. While the authors' slightly different setting justifies a new ablation study, the actual contribution remains limited. The third and primary claimed contribution is the introduction of a new dataset. However, the dataset is not adequately discussed, with minimal information provided. Given its central role in the work, the dataset requires a proper introduction and detailed description within the publication.

- The experimental protocol focuses on three specific tasks, while the model is intended to be generic. The authors should either (1) design a more comprehensive set of tasks across various domains to evaluate the model's generalizability or (2) focus on fewer tasks with more in-depth analysis to understand the specific genomic knowledge captured by the network. For example, the authors could investigate whether their model captures enhancer-promoter interactions better than existing models (see the insightful biological analyses in [1]) or if the attention maps can capture RNA secondary structures (see the recent work [2]). They could also leverage and adapt recent ideas from D3, a diffusion model for DNA sequences [3].

- Interpreting the performance differences between Evo, GenSLM, and GenomeOcean is challenging. Since all three models are trained with the same loss function and prompted similarly, the only differences lie in (1) their architecture and (2) pre-training data. The authors do not discuss the pre-training data or the extent of overlap between the downstream datasets and the pre-training datasets for each model, making it difficult to gain insights from the comparison.

While I acknowledge the high presentation quality, tutorial aspects, and strong engineering, the publication lacks sufficient contributions for a conference like ICLR. A major concern is the emphasis on the new pre-training dataset, which is not adequately discussed.

**Questions:**

Questions / Comments:

- The term "genome foundation model" is potentially confusing, as it implies the model can generate complete genomes. The authors should consider using the more common terms "sequence model" or "DNA sequence model".

- In Figure 3 (right), presenting losses for CLM and MLM on the same graph may be misleading, as losses across different training objectives cannot be directly compared.

- In Figure 3 (left and center), could the authors clarify the difference between "Transformer (MLM)" and "DNABert2" and explain the observed performance difference?

- Regarding the species generation experiment, the authors mention using 2kbp prompts. It is unclear whether this length provides sufficient context for species discrimination. Notably, HyenaDNA showed that 1kbp is insufficient for species differentiation and that at least 32kbp is required. While the species studied here might differ (the studied species here are not explicitly stated), further clarification on this point is needed.

- The authors should reference [4] in the context of generating species-conditioned sequences.



[1] https://www.biorxiv.org/content/10.1101/2023.08.30.555582v1.full.pdf

[2] https://www.biorxiv.org/content/10.1101/2024.07.27.605418v1.full.pdf

[3] https://www.biorxiv.org/content/10.1101/2024.05.23.595630v1.abstract

[4] https://link.springer.com/article/10.1186/s13059-024-03221-x

---

### Meta-Review · Area_Chair_QDGU · 2024-12-31

**Metareview:**

The type of model proposed in this paper can be valuable in principle. However, the reviewers have several serious concerns about the experimental evaluation (see the reviews for more details), and there is no author response. Given this, I am recommending rejection.

**Additional Comments On Reviewer Discussion:**

The authors have not submitted any rebuttal comments.

---

### Decision · Program_Chairs · 2025-01-22

Reject